# The Multi-Dimensional Biomarker Landscape in Cancer Immunotherapy

**DOI:** 10.3390/ijms23147839

**Published:** 2022-07-16

**Authors:** Jing Yi Lee, Bavani Kannan, Boon Yee Lim, Zhimei Li, Abner Herbert Lim, Jui Wan Loh, Tun Kiat Ko, Cedric Chuan-Young Ng, Jason Yongsheng Chan

**Affiliations:** 1Cancer Discovery Hub, National Cancer Centre Singapore, Singapore 169610, Singapore; lee.jing.yi@nccs.com.sg (J.Y.L.); bavani.kannan@nccs.com.sg (B.K.); lim.boon.yee@nccs.com.sg (B.Y.L.); li.zhimei@nccs.com.sg (Z.L.); abner.lim.m.s@nccs.com.sg (A.H.L.); loh.jui.wan@nccs.com.sg (J.W.L.); ko.tun.kiat@nccs.com.sg (T.K.K.); cedric.ng.c.y@nccs.com.sg (C.C.-Y.N.); 2Oncology Academic Clinical Program, Duke-NUS Medical School, Singapore 169857, Singapore; 3Division of Medical Oncology, National Cancer Centre Singapore, Singapore 169610, Singapore

**Keywords:** tumor mutational burden, microsatellite instability, multiomics, single cell transcriptomics, spatial transcriptomics

## Abstract

The field of immuno-oncology is now at the forefront of cancer care and is rapidly evolving. The immune checkpoint blockade has been demonstrated to restore antitumor responses in several cancer types. However, durable responses can be observed only in a subset of patients, highlighting the importance of investigating the tumor microenvironment (TME) and cellular heterogeneity to define the phenotypes that contribute to resistance as opposed to those that confer susceptibility to immune surveillance and immunotherapy. In this review, we summarize how some of the most widely used conventional technologies and biomarkers may be useful for the purpose of predicting immunotherapy outcomes in patients, and discuss their shortcomings. We also provide an overview of how emerging single-cell spatial omics may be applied to further advance our understanding of the interactions within the TME, and how these technologies help to deliver important new insights into biomarker discovery to improve the prediction of patient response.

## 1. Introduction

Recent advances in the field of immuno-oncology have led to a paradigm shift in the standard of care for human cancers. A better understanding of immune surveillance in which innate immune cells eliminate cancer cells, coupled with the discovery of T-cell immune checkpoint inhibitors (ICIs), have significantly improved survival outcomes and quality of life for many patients. Antibody blockade of immune checkpoints Cytotoxic T Lymphocyte Antigen 4 (CTLA-4) and Programmed cell Death 1 (PD1)/PD1 ligand 1 (PD-L1) have been shown to restore antitumor immunity in multiple tumor types such as melanoma [1,2], renal cell carcinoma [3,4], non-small cell lung carcinoma [5,6] and Hodgkin’s lymphoma [7,8]. However, as more tumor types demonstrating potential benefit to ICIs are noted, durable responses are observed only in small subsets of patients, whereas the large majority remains unresponsive [9,10]. ICIs can also result in immune-related adverse drug reactions as well as tumor hyperprogression [11,12]. There is thus an unmet need to identify better predictive biomarkers of response to ICIs to prescribe them in a more selective manner and to better understand mechanisms of therapeutic resistance.

Over the years, the tumor microenvironment (TME) has been increasingly studied for its key role in shaping immunotherapy response, of which the mechanisms involved are not yet fully understood [13,14,15]. The TME comprises a variety of non-malignant resident and infiltrating host cells, secreting factors and extracellular matrix (ECM) proteins surrounding the tumor, and may differ between cancers even of the same histological origin. The cross-talk between the TME components and the tumor involves the exchange of molecules such as cytokines, chemokines and mitogens, which, in turn, exert a profound influence on tumor initiation, progression and metastasis [16]. As such, there is a need to develop strategies to characterize the composition, function, activity and spatial location of cellular components in the TME to better understand the phenotypes that contribute to a tumor-favoring microenvironment as opposed to those that confer susceptibility to immune surveillance and thus immunotherapy.

In this review, we provide an overview of the emerging predictive biomarkers with a focus on the use of current and evolving technologies and models to study the TME at a profound level. We begin by examining conventional markers such as PD-L1 immunohistochemical expression, tumor mutation burden and DNA microsatellite instability, and discuss their clinical utility in detail. This is followed by a discussion of how gene expression profiling of the tumor microenvironment and immune landscape may be applied to understanding immunotherapy response. Lastly, emerging literature on new single cell and spatial transcriptomic technologies is presented. We summarize how these methods may be deployed to identify mechanisms that confer resistance to therapy, predict therapeutic response, and accelerate the identification of novel treatment targets.

## 2. Beyond PD-L1 Immunohistochemistry

Immune checkpoint inhibitors against the PD-1/PD-L1 axis have certainly come of age, being approved for clinical use across all tumor types in specific molecular contexts [17]. The PD-1 cell surface receptor is expressed on tumor-infiltrating immune cells and its ligand, PD-L1, is expressed on antigen-presenting cells and tumor cells, forming an interacting immune checkpoint axis that blocks the adaptive immune response against tumor cells and thereby enabling immune evasion [18]. Antibody-based therapy against PD-1 (e.g., pembrolizumab, nivolumab, sintilimab) and PD-L1 (e.g., atezolizumab, avelumab, durvalumab) now represents part of standard care in the oncology clinic, though response rates vary across cancer types and the ideal biomarker to guide precision therapy is still lacking. The use of PD-L1 expression as a biomarker of response to immune checkpoint inhibition has recently been reviewed elsewhere [19] and will not be extensively discussed here. Nevertheless, we herein summarize the recognized indications till date and examine how its use has evolved in the modern setting.

PD-L1 immunohistochemistry represents one of the earliest predictive assays developed to guide patient selection for checkpoint immunotherapy. However, despite early indications of PD-L1 immunohistochemistry as a promising tumor agnostic biomarker that is affordable and accessible, several problems began to surface including the need for different assay systems depending on the specific agent selected and cancer type. Variability in cut-off points, antibody performance, inter-user and inter-assay inconsistencies led to technical challenges in incorporating PD-L1 immunohistochemistry as a “one-size-fits-all” biomarker applicable across the entire cancer and immune checkpoint inhibitor conglomerate.

Four of such PD-L1 immunohistochemistry assays are currently recognized by the United States Food and Drug Administration (US FDA) as companion diagnostics: (1) Dako 22C3 for pembrolizumab for a range of solid tumors [20,21,22,23,24]; (2) Dako 28-8 for ipilimumab and nivolumab in non-small-cell lung cancer (NSCLC) [25]; (3) Ventana SP142, and (4) Ventana SP263 for atezolizumab in urothelial carcinoma, triple-negative breast cancer or NSCLC [26,27,28,29]. Depending on the specific tumor context and assay, a range of cut-offs exists for PD-L1 positivity calculated based on tumor and/or tumor-infiltrating immune cell expression. Clearly, this gives rise to reproducibility issues and risks discordant results amongst pathologists [30,31]. Adding to the complexity, the relative importance of each immune cell compartment (e.g., macrophage or lymphocyte subsets) remains an active area of research [32], and further optimizations of scoring algorithms are required. The implications of specific genomic alterations of the PD-L1 gene, such as amplifications [33,34] and structural variations [35,36] also require validation.

Meanwhile, ongoing efforts to overcome issues of assay sensitivity and improve reproducibility include the use of multiplex immunohistochemistry/immunofluorescence coupled with automated digital image analysis [37,38]. As current standardized methods to using PD-L1 immunohistochemical expression continue to mature in a fit-for-purpose and cancer-specific manner, we are simultaneously witnessing the development of novel multiplexed approaches to interrogate the tumor and its microenvironment at a multi-dimensional scale to complement PD-L1 as a biomarker.

## 3. Tumor Mutational Burden—Universal Biomarker, or Not Quite?

Malignancies are consequences of genomic alterations encoding for mutant proteins. Following proteosomal degradation, peptides originating from mutant proteins with higher affinity for major histocompatibility complex (MHC) molecules than normal peptides are presented by MHC molecules as tumor neoantigens on the cancer cell surface. The recognition of these tumor neoantigens by host T cells is one of the key factors in determining immunotherapy response. Tumors harboring high total number of mutations per coding genomic region analyzed (mut/Mb), also denoted as high tumor mutational burden (TMB), have an increased level of tumor neoantigens and present higher immunogenicity, thereby reducing the probability of immune escape of tumor cells [39,40]. Additional related factors include the host MHC and T cell receptor landscape [41,42]. Supporting this assumption, viral-associated cancers, such as Merkel cell carcinoma and Kaposi sarcoma, have demonstrated higher than expected response rates to ICIs, attributed to the presentation of viral antigenic peptides enabling enhanced tumor immunogenicity [43,44].

Early studies on various cancers showed that patients with high TMB (TMB-H), calculated using a comprehensive genomic profiling assay, were more responsive to ICIs than patients with low TMB, highlighting TMB as a promising predictive biomarker [45,46,47]. Subsequent studies provided incremental evidence for improved ICI response in TMB-H tumors, especially melanoma and lung cancer [11,48,49,50,51,52,53,54,55,56]. In June 2020, pembrolizumab, an anti-PD1 immune checkpoint inhibitor, was approved by the US FDA for treatment of patients with TMB-high (TMB-H) tumors (≥10 mut/Mb), following a retrospective analysis of 10 cohorts of patients with previously treated TMB-H solid tumors in the phase II KEYNOTE-158 trial [54]. It is critical to note that TMB was determined by a specific approach using the 324 gene F1CDx targeted panel, and that major tumor types, such as prostate cancer, hormone receptor-positive breast cancer and microsatellite stable colorectal cancer, were not included in this study. The most appropriate method for calculating TMB, the optimal cut-off for defining TMB-H, and the specific tumor context are some of the important issues that remain to be determined. Recently, in a study of over 10,000 patients across 31 cancer types from The Cancer Genome Atlas (TCGA), the classification of tumors as TMB-H was shown to vary in a cancer type-specific manner. In particular, the utility of TMB must be interpreted based on whether CD8 T-cell infiltration is positively-correlated with neoantigen load [57]. Importantly, TMB-H failed to predict ICI response in patients with multiple cancer types including glioma, prostate cancer, breast cancer, and other tumor types where neoantigen load is not associated with CD8 T-cell infiltration.

In the early days of exploring TMB as a potential predictive biomarker, whole exome sequencing (WES) of tumors was typically carried out to determine mutation load, and is still considered the gold standard approach. In more recent times, precision oncology platforms now utilize next generation sequencing (NGS) of targeted panels such as the FoundationOne CDx (F1CDx) [58], Memorial Sloan Kettering Cancer Center’s Integrated Mutation Profiling of Actionable Cancer Targets (MSK-IMPACT) [59], and FoundationACT (Assay for Circulation Tumor DNA) [60] assays which are significantly more cost-effective, making it more feasible to be routinely implemented in the clinical setting. Still, the concordance of various DNA sequencing approaches remains a technical challenge and requires further optimization and validation. The correlation of TMB with treatment response and survival outcomes of patients treated with ICI remain inconsistent across cancer histologies, with some subtypes, such as gliomas demonstrating the opposite correlation [61]. These variable responses to ICI together with the lack of a standardized approach to determine TMB pose as hindrances to the implementation of TMB as a bone fide tumor agnostic predictive biomarker (reviewed extensively in [62,63]. As efforts are ongoing to standardize the reporting of TMB measurements across the various NGS-based panels to help achieve consistency in findings [64,65], it is likely that a further step has to be taken to integrate other orthogonal biomarkers of ICI response, as is in subsequent Section 4, Section 5 and Section 6.

## 4. DNA Mismatch Repair and Microsatellite Instability

Microsatellite instability (MSI) is a molecular feature caused by defective DNA mismatch repair (dMMR). Attributed to either germline mutations or acquired somatic alterations in one of the MMR genes (*MLH1, MSH2, MSH6, PMS2*), cells with dMMR proteins are unable to rectify errors during DNA replication such as base substitution mismatch, frameshift mutation and slippage, resulting in high levels of MSI (MSI-H), accumulation of numerous mutations favoring malignancies and increased neoantigen burden. In most cases, this leads to an increase in TMB and immunogenic neoantigens, rendering MSI-H/dMMR tumors potentially susceptible to immunotherapy [66,67]. MSI-H/dMMR signatures are found in a small proportion (<5%) of all cancers, most commonly colorectal, endometrial and gastric adenocarcinomas [68]. In keeping with these findings, the efficacy of ICIs has been demonstrated in MSI-H/dMMR colorectal cancers [69,70,71,72,73], endometrial cancers [74,75] and gastric cancers [76]. In 2017, the US FDA approved pembrolizumab for treatment of unresectable or metastatic solid tumors from any histology that exhibit MSI-H/dMMR [77].

While certainly a potentially useful predictive biomarker for ICI therapy in the clinic, MSI-H/dMMR signatures remain uncommon across all tumor types, and responses are not universal. Differences in the TMB, neoantigen load, and infiltrating lymphocytes in these tumors probably contribute to some of the variable responses observed [47,71,78]. In addition, the optimal testing methods across the tumor spectrum remain to be defined. As with TMB, the lack of standardization in microsatellite panels and thresholds used has caused great variations in reports on MSI-H. The low frequency of MSI-H tumors involved in clinical studies should also be considered when determining its potential as a predictive biomarker across cancer types. For example, despite observing higher response rates in MSI-H tumors in a recent phase 2 clinical trial in gastric and gastroesophageal junction cancer patients, the sample number of MSI-H tumors was very low in the cohort (4% of the participants) and most responses were observed in non-MSI-H patients [79]. Current approved tests include the assessment of MMR protein expression by immunohistochemistry, pentaplex polymerase chain reaction for MSI, as well as NGS-based approaches. In particular, NGS-based approaches may provide both the mutational events and reveal unique mutational signatures [80,81]. These broad profiling strategies may reveal additional mutagenic signatures leading to high TMB, including alterations in other DNA damage response and repair pathway genes [82], or the presence of mutagenic carcinogens [83,84,85,86]. The absence of MSI-H/dMMR signatures should not preclude the consideration for ICI therapy, and a compendium of relevant biomarkers should be included in the assessment.

## 5. Gene Expression Profiling of the Tumor Microenvironment and Immune Landscape

Molecular profiles of the tumor and its surrounding immune components within the TME possess tremendous value in serving as prognostic and predictive biomarkers. Specifically, Gene Expression Profiling (GEP) has been widely performed of most cancer types and has contributed to remarkable progress in identifying immune-related gene expression-based signatures for risk stratification and predicting clinical outcomes [87,88,89,90], including for predicting response to immunotherapy [91].

Earlier, we discussed the pivotal yet imperfect role of tumor genetic markers, including TMB and MSI, as predictive biomarkers for ICI therapy. More recently, a T cell-inflamed GEP signature containing interferon-gamma responsive genes has been demonstrated to correlate with clinical benefit to PD-1-directed immune checkpoint blockade. Tumors expressing the signature indicate the presence of immune cell infiltration and evasion of immune surveillance [92,93]. A final Tumor Inflammation Signature profile containing 18 genes (TIS) was shown to correlate with response to pembrolizumab, and may outperform PD-L1 immunohistochemistry [93,94,95,96]. Expression of TIS genes has highly conserved co-expression patterns independent of tumor origin, and subsets of patients with elevated TIS scores can be identified across all tumors [97,98]. Further evaluation of more than 300 patient samples across 22 tumor types from four clinical trials showed that TMB and a T cell-inflamed GEP independently identified responders and non-responders to pembrolizumab [99]. Apart from T cell signatures, B cell-related gene signatures have also been shown to predict immunotherapy response in melanomas and sarcomas [100,101,102,103,104]. Taken together, these data support the combined use of PD-L1 immunohistochemistry, TMB and GEP in the prediction of ICI benefit (Figure 1).

PD-L1 immunohistochemistry (IHC) is one of the earliest predictive assays developed to guide patient selection for checkpoint immunotherapy. It represents an affordable and quick way to detect PD-L1 expression in tumor tissue sections, and therefore enable selection of patients for ICI therapies based on PD-L1 expression levels. However, due to its irreproducibility, more advanced and robust methods, such as gene expression profiling using microarray, bulk RNA-seq and NanoString profiling, have vastly taken precedence over PD-L1 IHC. The correlation of gene expression levels with ICI responses has led to the discoveries of novel potential predictive biomarkers. TMB and MSI are genomic signatures that have been widely explored for their use as predictive biomarkers. High TMB and MSI levels have been shown to positively correlate with ICI responses, but these results have been challenged in studies that report otherwise.

## 6. Evolving Platforms for Gene Expression Profiling

Traditionally, hybridization-based microarray was commonly employed for gene expression profiling and immune cell deconvolution. Probes corresponding to a set of pre-defined mRNAs of known sequences are spotted onto a microarray slide, in which complementary DNA (cDNA) reverse-transcribed from RNA would hybridize to [105,106]. The microarray technology can be used to identify transcriptional differences between populations of interest; for example, to identify sensitive and resistant patient populations to therapy. However, owing to the limitations of this technique such as requiring prior knowledge of the target sequences, requirement of larger amounts of starting material, lack of standardization pipelines to integrate datasets from various sources, and challenges in data reproducibility in repeated experiments due to technical and sample variations, microarrays are no longer widely utilized in contemporary gene expression studies [107,108]. Rather, next generation RNA sequencing (RNA-seq) has largely replaced microarrays for gene expression profiling. In contrast to microarrays, there is no need for prior knowledge of the sequences of interest when performing RNA-seq, which is capable of distinguishing and quantifying known and novel genes across the whole transcriptome [109,110]. With the advancements of computational methods and bioinformatics algorithms, the data retrieved from RNA-seq of bulk tissues can be effectively analyzed to greater depth and provide crucial information of the TME and the role it plays in influencing immune contexture in cancer [111]. The analytic workflow from RNA-seq typically includes Differential Gene Expression (DGE) to determine genes that are differentially-expressed under various conditions. DGE is routinely complemented with Gene Set Enrichment Analysis (GSEA), which may identify signals emanating from relevant tumor subgroups (e.g., responsive or resistant to therapy) [112].

What is noteworthy is that the bioinformatic tools widely available now can provide higher level information of the full transcriptome from bulk RNA-seq data, creating opportunities for scientists to uncover new information about the TME without the need to curate a list of genes of interest beforehand. Although the data footprint of RNA-seq is significantly larger than that of microarrays, and thus require longer computational time and storage space, RNA-seq provides a more comprehensive picture of the transcriptome. Additionally, computational research is a rapidly evolving field, constantly discovering new algorithms and ways to interpret RNA-seq data for the understanding of the TME and enhancing the applicability of RNA-seq in the advancements of immunotherapy. As an example, there are currently more than 50 deconvolution algorithms for RNA-seq, such as CIBERSORT, TIMER, EPIC and quanTIseq [113,114], enabling the rapid identification and quantification of the various subpopulations of immune and other cells infiltrating the tumor. Weighted Gene Co-Expression Network Analysis (WGCNA) can be further applied to pinpoint novel transcription factors that are greatly associated with the infiltrated tumor-associated immune cells [115]. From there, gene expression networks composed of the identified transcription factors and immune-related genes can be constructed and validated by TCGA cancer data to correlate gene expression data to tumorigenesis and patient survival [116,117]. The application of such bioanalytical tools has certainly shed light on the influence of TME components in response to immunotherapy.

The availability of a comprehensive and rapidly deployable assay of the immune cell population and relevant GEP signatures would certainly be useful in the oncology clinic. For example, the TIS GEP has been incorporated into a 770-gene PanCancer immune profiling panel on the NanoString nCounter platform, which is able to detect and quantify multiplexed gene expression from samples, including limited or degraded analytes isolated from formalin-fixed paraffin-embedded (FFPE) tissues. The system involves a direct overnight hybridization of unique fluorescent color-coded barcodes onto specific mRNA targets of interest and subsequent tabulation of the raw counts of genes expressed onto a precise digital data file. The set of color-coded barcodes make up a customizable gene expression panel, which allows users to study a specific set of genes according to their experimental interests [118]. The association between tumor immune-related genes and various clinical outcomes has been investigated using the NanoString nCounter system in lung cancer [119], breast cancer [120], bladder cancer [121], leukemia [122], hypopharyngeal carcinoma [123] and angiosarcoma [85,124].

## 7. Emergence of Single-Cell Technologies

Single-cell sequencing has overcome the limitations of bulk sequencing, which measures the averaged gene expressions from a heterogeneous population of cells. The need to distinguish tumor-to-tumor and cell-to-cell differences for determining how tumor heterogeneity influences patient responses to treatment, has led to the rapid development of many single-cell sequencing technologies. With single-cell sequencing, researchers can now sequence individual cells to profile their genomic, transcriptomic, and other “omic” information. Consequently, single-cell sequencing enables the investigation of the transcripts and mutations detected in individual genes at the cellular level, allowing the discovery of complex and rare populations as well as the roles they play in the TME [125,126].

Single cell RNA-seq (scRNA-seq) technology enables high throughput transcriptomic profiling of individual cells. First published in 2009, scRNA-seq has evolved massively over the past decade, giving rise to more streamlined and affordable scRNA-seq technologies such as microfluidic-based microwell-based, droplet-based and in situ barcoding [127,128]. These methods, however, require cells or nuclei to be in suspension, leading to the loss of information on the spatial contexture. The standard workflow of scRNA-seq involves single cell isolation, cell permeabilization for release of RNA, reverse transcription to cDNA, cDNA amplification, library preparation and sequencing. For example, the 10X Genomics Chromium system is one of the most conventional platforms used in single-cell transcriptomic sequencing. It utilizes a microdroplet-based system to measure transcripts from individual cells, allowing the analysis of thousands of cells at the same time. Briefly, the workflow includes sample preparation, library construction, data sequencing, data analysis, and data visualization [129]. A more comprehensive and customized analysis can be performed using Seurat [130], which is an R package compatible with most of the algorithms and tools developed for single-cell transcriptomic analysis. It allows data filtering, normalization, scaling, optional imputation and batch effect removal, dimensional reduction, clustering, as well as visualization to take place in an integrated analysis.

In immuno-oncology, single-cell approaches have provided an opportunity for high-dimensional characterization of cancers across their multi-omic landscapes beyond bulk genomic and transcriptomic analyses [131,132], enabling the identification of rare cell subpopulations within the diverse TME that may determine sensitivity to ICI therapy in specific cancers [133,134,135,136,137,138,139]. More recently, a novel pan-cancer “stemness” gene expression signature has been evaluated at the single-cell level using CytoTRACE and was suggested to confer resistance to ICI treatment perhaps in a tumor agnostic manner [140,141]. In the same light, the pan-cancer identification of T-cell subsets in distinct transcriptional states may lead to an immune-based stratification of patients. In particular, the relative abundance of tumor-reactive CD8+ cytotoxic T-cells with a terminal exhaustion phenotype and CD4+ regulatory T-cells expressing TNFRSF9 may modulate responses to immunotherapy [142]. The integration of single cell T-cell receptor (TCR) sequencing has also identified shared gene expression profiles of neoantigen-specific CD8+ and CD4+ T-cells within metastatic human cancers, which may provide useful biomarkers for immunotherapy as well [143].

Apart from high resolution interrogation of the transcriptome, current methods are capable of whole genomic sequencing of single cells [144]. The Di-Electro-Phoretic Array (DEPArray) system is a microchip-based digital sorter combining precise microfluidics and microelectronics in a highly automated platform, allowing the isolation of single cells for downstream genomic amplification and sequencing [145]. Its application, though limited in terms of scalability and high costs involved, enables the recovery of individual rare cells of interest from heterogeneous samples for detailed analysis [146]. A more cost-efficient strategy for single cell genomic analysis is to use a targeted sequencing approach for the detection of specific gene mutations and clonal structure within the tumor, achievable using the Tapestri system [147,148,149]. These technologies have been applied already in the context of mutation tracking in blood cancers, and could be useful in immunotherapy research in due time. We anticipate that as single-cell platforms and data mature, we expect greater implementation of these technologies alongside conventional clinicopathological and bulk “omic” assays to better inform a precision immunotherapy strategy in the clinic.

## 8. Spatial Transcriptomic Technologies

scRNA-seq has significantly advanced our understanding of the different cell types that co-exists in a tumor. However, scRNA-seq studies are not able to provide information on the spatial organization of the different cellular constituents of a tumor because a typical scRNA-seq methodology involves physical disruption of a tumor to obtain single cells for subsequent analysis. Spatial transcriptomic technologies (STTs) utilize on-site barcoding methods that ligate specific barcode sequences that provide information on both the location where a population of mRNAs is isolated, as well as the standard molecular barcoding information for each ligated mRNA molecule. These data can then be digitally merged to provide spatial and gene expression information. Several multiplexed in situ hybridization approaches (such as MERFISH, SEQFISH, SABERFISH) [150,151,152] have been developed, and are able to image and unambiguously distinguish thousands of densely packed RNA species within a cell. These techniques have come up with various solutions to barcode the numerous mRNA species with reduced recognition error rate and can generate single molecule resolution maps of tens to thousands of pre-selected RNA species at subcellular level. Due to inherent technical complexity, time-consuming processing and low throughput, these techniques have become less attractive compared to other STTs such as Visium and GeoMX Digital Spatial Profiling (DSP). 

The 10X Genomics Visium spatial gene expression platform consists of a customized microscope slide that contains four capture regions for mounting tissues (frozen or FFPE). The dimension of each region is 6.5 mm by 6.5 mm. Each square region is printed with 4992 spots and each spot has a diameter of 55 µm that contains an array of DNA-based capture probes. Each capture probe contains a unique spot-specific spatial barcode, a unique molecular identifier as well as a poly (dT) tract that is used to capture mRNA containing poly (A) tail and potentially enables the profiling of the whole poly (dA)-tailed transcriptomes [153]. The workflow involves standard tissue-fixing and staining (haematoxylin & eosin or immunofluorescence) that is followed by image capture. Subsequently, the tissue is permeabilized to release the mRNAs that are then captured by the poly (dT) tract of the nearest capture probes. The captured mRNAs are then reverse transcribed to cDNAs that have both the unique spatial barcode, hence the “address” of where the mRNAs originate from, as well as the unique molecular barcode from the corresponding capture probes. The cDNA libraries generated can then be pooled for subsequent standard NGS. The gene expression information obtained can be mapped to specific locations within the tissue by aligning the image captured earlier in the workflow to known spatial barcode information. The Visium system has been reported to be compatible with FFPE samples [154]. Despite a significant correlation between FFPE and frozen samples in terms of their levels of gene expression, FFPE samples generated less unique genes and unique mRNA [154]. Currently, the gene expression data obtained from Visium cannot be resolved to single-cell level. That is because each spot, with a diameter of 55 µm, would be able to fit in approximately 3–10 cells. However, it appears that it is possible to computationally integrate and enhance the resolution of the captured regions [155].

Nanostring GeoMx DSP can analyze spatial expression of preselected panels of probes that target specific RNAs or proteins within regions of interest (ROIs) in a tissue section that is prepared from either fresh frozen or FFPE [156,157]. This technique involves the use of either gene-specific probes, with photocleavable molecular barcodes, that hybridize to specific RNA species or for protein detection, using antibodies that are tagged with specific molecular barcodes. After probe hybridization or protein detection by tagged antibodies, the tissue is stained with the appropriate antibodies conjugated with fluorescent probes to reveal cellular morphology and markers which guide the subsequent ROI selection. Each ROI is individually exposed to UV light to release the photocleavable barcodes from the gene-specific probes or the tagged antibodies. The cleaved barcodes are subsequently aspirated, and can be identified with the either the NanoString nCounter system or standard NGS. Due to its detection limit, the selected ROI should have at least 200 cells. However, in some cases where probes only target highly expressed RNAs, as few as 21 cells are required per ROI [156]. The specimen placement area (14.6 mm × 36.2 mm) allows the use of most tissue sections mounted, based on standard methods, on regular microscope slides. Since gene expression is mapped back to a specific ROI rather than to a precise coordinate within the tissue, the spatial resolution for the gene expression data is affected by the size and shape of the chosen ROI.

Recently, new STTs based on solid phase-bound capture probes with on-site barcoding capability have emerged that potentially can provide higher resolution (at single cell and subcellular scale), than the 10X Genomics Visium system. These STTs are able to utilize beads or spots that are smaller compared to Visium. These include Slide-seq and its improved version Slide-seqV2 (spot size: 10 µm) [158,159], High Definition Spatial Transcriptomics (HDST; spot size: 2 µm) [160] and DBiT-seq (Deterministic Barcoding in Tissue for spatial omics sequencing; spot size: 10 µm) [161]. The latest in this category is STEREO-seq (SpaTial Enhanced REsolution Omics-sequencing), which has sub-cellular resolution as well as increased capture region to accommodate bigger-sized tissue (available option for capture region: 50, 100 & 200 mm^2^). STEREO-seq utilizes DNA nanoball (DNB) patterned array chip that contains capture probes on spots with a diameter of about 220 nm with centre-to-centre distance of either 500 nm or 715 nm. Given the average size of a cell is about 100 µm^2^, the coverage per cell is approximately 400 spots, which results in detailed sub-cellular resolution that is on par with multiplexed FISH. Therefore, STEREO-seq can potentially bridge the gap between scRNA-seq and STTs. Taken together, these new spatial technologies result in better characterization of the functions and structural aspects of a tumor and its microenvironment [162].

## 9. Application of STT in Interrogating the Tumor Immune Landscape and Tumor Microenvironment

A recent study on the interactions between the tumor and its microenvironment showed that prostate cancer cells have different transcriptional signatures which are dependent on their location within a tumor. Furthermore, the study was able to detect different transcriptional signatures in different parts of the tumor that are important in metabolism, proliferation, and inflammation [163]. In a separate study, the metastatic melanoma microenvironment was found to contain different cancer cell populations with distinct melanoma expression signatures. Furthermore, the lymphoid cells found adjacent to melanoma cells were transcriptionally different to those found further away from these melanoma cells [164].

Currently, there are various methods for integrating scRNA-seq with spatially resolved transcriptomics to identify, characterize, and localize rare or unique group of cells within a tumor. One study into pancreatic ductal adenocarcinoma (PDAC) used scRNA-seq data to identify various cellular constituents, which include cancer cells, ductal epithelial cells, immune cells and fibroblasts. STT was then used to show the locations of these different cell types, and thus provided information on the interaction between immune cells and PDAC cells [165]. Another recent study on cervical squamous cell carcinoma (CSCC) combined data from single nuclei RNA sequencing (snRNA-seq) from frozen tumors with data from STEREO-seq on both CSCC and normal cervical tissues to identify the different cell types and to delineate the spatial interactions between cancer cells and their tumor microenvironment [166]. This study showed that a group of cancer-associated fibroblasts (CAFs) promoted cancer growth and metastasis by preventing lymphocytes infiltration and changing the tumor extracellular matrix. This study also provided insights into the relationship between HPV infection with cancer metabolism and immune response.

STTs can be used to study different niche compartments within a tumor. Furthermore, such studies can reveal the cellular constituent and the individual cell function by delineating their respective transcription profiles. Ji et al. used a multi-omics approach, which include spatially-resolved transcriptomics, to study cutaneous squamous cell carcinomas and the niches found within. At the leading edges of tumors they found a group of tumor-associated keratinocytes that resided in fibrovascular niches that are enriched for immunosuppressive Treg cells. Furthermore, these tumor-associated keratinocytes could promote cancer progression by increasing the malignant potential of adjacent normal cells [167].

STTs can be applied to identify the different tumor clones and their respective transcriptomes. One can then identify important tumor-specific pathways that are potentially druggable. Wang et al. used STT to assess metabolic networks in prostate cancer. Based on computer modelling of the spatially-resolved transcriptomics data, they identified tumor-specific metabolic pathways that could be targeted by small molecule inhibitors [168]. In studies where pre-treatment and post-treatment samples were available, STT can provide valuable data on treatment response [169,170]. Hwang et al. compared PDAC specimens from treatment-naïve versus neoadjuvant-treated patients. They found that basal-like PDAC cell transcriptome promoted relatively more immune cell infiltration when compared to those of classical-like PDAC cells. Additionally, the immune cell infiltrates associated with basal-like PDAC cells were enriched for lymphoid cells and were distinct from those associated with classical-like PDAC cells that were enriched for macrophages. Therefore, this suggested that the therapeutic strategy for basal-like PDAC are immune checkpoint inhibitors, while for classical-like PDAC cells the suggested therapies include myeloid-directed therapies such as CD40 agonists and Transforming Growth Factor β (TGF-β) modulators (e.g., losartan) [170]. GeoMX DSP has also been used to identify sensitivity to checkpoint inhibitors. One study into melanoma showed that PD-L1 expression in macrophages, but not melanoma cells, was the determining factor for treatment responses, overall survival and progression-free survival in patients [171]. In another study, NSCLC tumor samples were used to study the interaction between cancer cells and other cells in the TME. The study found biomarkers that predicted response to ICIs [172]. Cells with high CD56 expression correlated with improved survival while those with high CD127 expression found within the tumor compartment were found to be resistant to immunotherapy [172].

## 10. Conclusions and Prospects

The field of immuno-oncology is progressing at an accelerating pace, and potential predictive biomarkers are rapidly emerging. Beyond profiling of tumor characteristics, information derived from systemic factors including the gut microbiome [173,174,175], as well as host immune response and other circulating analytes [176,177], add further layers of complexity. Assays using tumor organoid models [178], humanized mouse models [179], and ex vivo tumor fragment platforms [180] may provide a dynamic means to simulate the host-tumor ecosystem and forecast real-life responses. Soon, it is highly anticipated that single cell multi-omic sequencing technologies encompassing the genome, transcriptome, epigenome and proteome will provide the intricate tools required to dissect and characterize the complexities of the cancer cell. Combined with multi-dimensional information of the TME through spatial and temporal localization, the identification of highly specific biomarkers that define immunotherapy response in each individual patient will usher in a new era for precision immuno-oncology (Figure 2).

Multi-dimensional tools have emerged over the recent years to improve the resolution of TME analysis and potentiate discovery of novel predictive biomarkers. Three-dimensional models generated from tumor specimens may be used to mimic in vivo TME and predict real-life treatment responses. Single-cell technologies such as 10X Genomics and MissionBio Tapestri systems enable transcriptomic and genomic profiling at the single-cell level, allowing deconvolution of heterogenous cell populations. Spatial technologies, such as 10X Genomics Visium, NanoString GeoMx and STEREO-seq, enable the visualization of gene expression in the context of the tissue morphology. The multi-dimensional approach of studying the TME provides a comprehensive understanding of the TME, and may shape how immunotherapy may be tailored to maximise efficacy. Profiling of the gut current microbiome, immune factors, and other circulating analytes, may provide valuable information and complement current strategies in better predictions of ICI benefit.

## Figures and Tables

**Figure 1 ijms-23-07839-f001:**
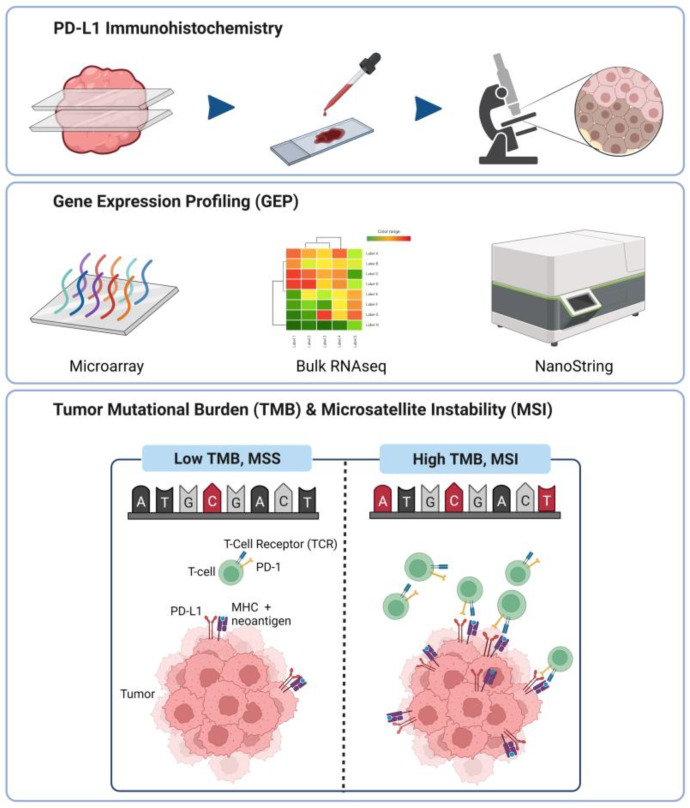
Conventional technologies used for prediction of ICI benefit in cancer patients.

**Figure 2 ijms-23-07839-f002:**
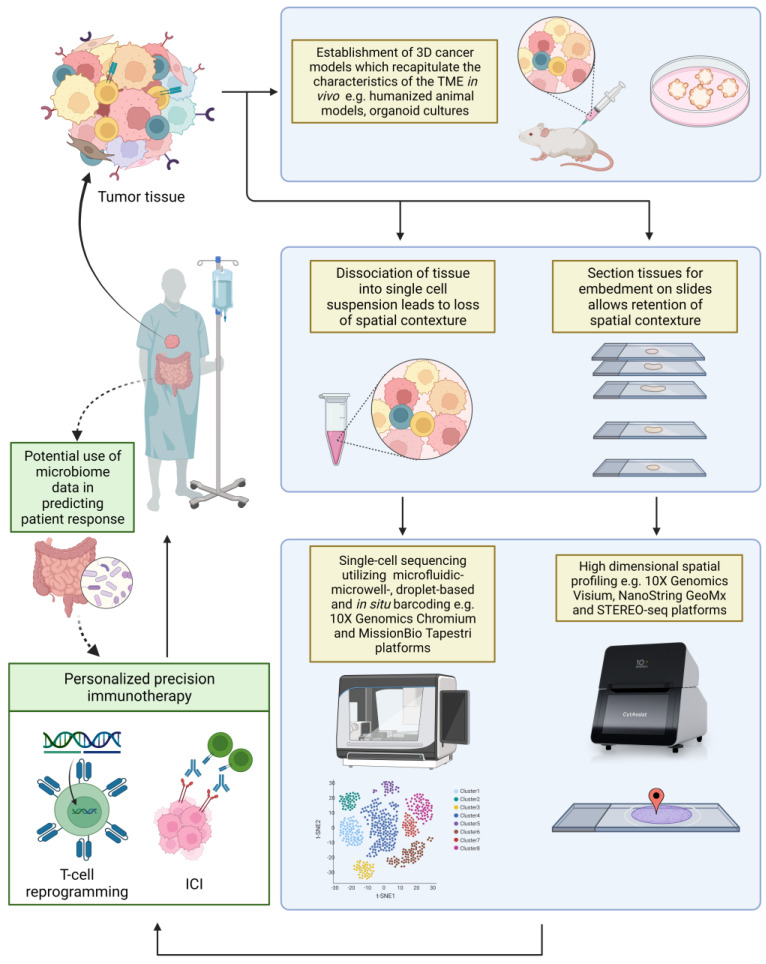
Emerging high-dimensional technologies in the field of immune-oncology.

## Data Availability

Not applicable.

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
