# Peer review of "The Multi-Dimensional Biomarker Landscape in Cancer Immunotherapy"

_ijms, 2022, doi:10.3390/ijms23147839_

Round 1
Reviewer 1 Report
The manuscript titled "The multi-dimensional biomarker landscape in ..." by Jing Yi Lee et al is a pretty good piece of paper that doesn't require too much revision before publication.
However, a few points need to be improved.
Below are my comments.
1. The first 2 sentences of the abstract for me to rewrite are kind of confusing.
2. Introduction - it is probably a bit too short, for such a work and the number of cited articles is probably a bit too little?
3. "In this Review, we ..." - this part needs to be rewritten - so far it is too general written. Please expand it.
4. "As current standardized methods to using ..." - what and what is this sentence for?
5. "Although efforts have been made to standardize the methodologies ..." please remove or redo this. I leave the decision to the authors.
6. Fig 1 - very sloppy, please correct it.
7. "Certainly, as single-cell platforms and data mature ..." - I don't think this sentence is complete either, especially in this form.
8. "Recently, new STTs based on solid phase-bound capture probes with on-site ..." - this entire paragraph is spooky, please edit it.
9. Fig 2 - also very sloppy. Please improve its quality.
Overall a very nice job, but requires a lot of work to slightly improve it.
I recommend major revision.
Reviewer 2 Report
The authors summary recent technology and biomarkers very well. This review will help readers of this journal to understand the meaning of stability and instability of several tumors.
Reviewer 3 Report
This review article is summarizing most widely used technologies and biomarkers as useful tools for predicting immunotherapy outcomes in patients and discusses their analysis to improve the prediction of patient responses.
This straightforward analytical methodology reported by the authors verified that total interactions within tumor microenvironment (TME) would help to deliver understanding of those specific interactions. The review article is concluded with a collection of 180 important and relevant references. Additionally, 2 (two) beautiful figures are very informative and with concise important sequential data analysis presentation.
This correlative analysis review will constitute the important goals and novelty of this paper.
The following suggested changes and recommendations should be introduced before the publication of the manuscript.
1. Title: Specific suggestion to replace “landscape” with “importance
2. Page 2, line 79, “consortium” should be replaced with “conglomerate”
3. Page 4, line 159, please explains, “which sections are subjects of this elaboration”?
4. Page 10, line 482, please insert the literature reference for “predicted response to ICls”
5. Page 11, conclusions, line 491. Please insert the literature reference for “humanized mouse models”
The manuscript is of very good quality and urgent importance and is comprehensively well written and edited in order to meet the standard for the review articles published in International Journal of Molecular Sciences. Thus, I certainly recommend it for publication after the correction of these suggested minor changes.

Round 2
Reviewer 1 Report
The authors have improved this manuscript very well. It seems to me that it can be accepted as it is.